# Reablation in Atrial Fibrillation Recurrence and Pulmonary Vein Reconnection: Cryoballoon versus Radiofrequency as Index Ablation Procedures

**DOI:** 10.3390/jcm11195862

**Published:** 2022-10-03

**Authors:** Ana Isabel Molina-Ramos, Amalio Ruiz-Salas, Carmen Medina-Palomo, Francisco Javier Pavón-Morón, Jorge Rodríguez-Capitán, Mario Gutiérrez-Bedmar, Germán Berteli-García, Ignacio Fernández-Lozano, Juan José Gómez-Doblas, Manuel Jiménez-Navarro, Javier Alzueta-Rodríguez, Alberto Barrera-Cordero

**Affiliations:** 1Unidad de Gestión Clínica del Corazón, Hospital Universitario Virgen de la Victoria, 29010 Málaga, Spain; 2IBIMA-Plataforma BIONAND and Universidad de Málaga, 29590 Málaga, Spain; 3Centro de Investigación Biomédica en Red de Enfermedades Cardiovasculares (CIBERCV), Instituto de Salud Carlos III, 28029 Madrid, Spain; 4Unidad de Arritmias, Unidad de Gestión Clínica del Corazón, Hospital Universitario Virgen de la Victoria, 29010 Málaga, Spain; 5Departamento de Medicina Preventiva y Salud Pública, Facultad de Medicina, Universidad de Málaga, 29010 Málaga, Spain; 6Unidad de Arritmias, Servicio de Cardiología, Hospital Universitario Puerta de Hierro, Majadahonda, 28222 Madrid, Spain

**Keywords:** atrial fibrillation, catheter ablation, recurrence, cryoballoon, radiofrequency, pulmonary vein reconnection, reablation

## Abstract

Pulmonary vein (PV) isolation is a well-established rhythm control therapy in atrial fibrillation (AF). Currently, there is no consensus on which ablation technique to use for the first procedure, cryoballoon (CB) or radiofrequency (RF). A retrospective cohort study was conducted on 1055 patients who underwent a first ablation, to assess both techniques based on the need for reablation. Patients with CB (*n* = 557) and RF (*n* = 498) ablations were clinically characterized and the need for reablation during a 30-month follow-up was used as the primary endpoint. Independent variables were analyzed to identify potential predictors. The need for reablation was significantly lower in the CB group than in the RF group (hazard ratio = 0.45 and 95% confident interval = 0.32–0.61; *p* < 0.001); in both paroxysmal and persistent AF, using a full-adjusted regression Cox model by age, sex, smoking, hypertension, diabetes mellitus, dyslipidemia, severe obstructive sleep apnea, dilated left atrium, persistent AF and early recurrence. RF ablation, dilated left atrium, persistent AF and early recurrence were identified as independent predictors of reablation. In addition, the CB-redo subgroup had a lower PV reconnection than the RF-redo subgroup. In conclusion, CB ablation suggests a reduction in the need for reablation and lower PV reconnection during the follow-up than RF ablation.

## 1. Introduction

Pulmonary vein (PV) isolation with catheter ablation is a well-established rhythm control treatment of atrial fibrillation (AF) and is currently one of the main arrhythmic ablated substrates. Although research in this area has grown exponentially in recent years, certain aspects remain open to debate.

Recent American and European AF guidelines have reviewed the indications for the management of AF and the ablation techniques, highlighting the importance of strictly controlling factors that may alter or increase the recurrence and the fundamental role of the patient, who should be correctly informed of the risks and benefits when making the final decision [1,2]. In the case of paroxysmal AF, the objective is complete PV isolation; however, the target is not so clear for persistent AF. Although previous studies have compared different ablation strategies, it has not been shown that adding more ablation lines in the first procedure brings the additional benefit of reducing the recurrence rate [2,3].

The two main techniques are single-shot cryoballoon (CB) ablation and point-by-point radiofrequency (RF) ablation. Previous studies have compared both methods for effectiveness and safety and conclude that CB is not inferior to RF for paroxysmal AF, and both show a similar rate of complications [4,5,6,7,8]. However, these studies have short follow-up periods (up to a maximum of 18 months) and most of them do not include patients with persistent AF. Currently, there is no consensus about the preferred ablation technique to use, and the choice is left to the experience of the electrophysiologist, the availability of the center/hospital resources and the preference of the patient.

The purpose of our study was to compare patients who underwent CB and RF ablation procedures for AF during a 30-month follow-up using demographics, cardiovascular risk factors and clinical variables. Different Cox regression models were used to assess the need for repeat ablation because of arrhythmia recurrence during the follow-up in both groups of patients, as the primary endpoint. The independent variables in these multivariable models were explored as potential predictor variables of reablation. Finally, the PV reconnection was examined as the main arrhythmogenic mechanism involved in recurrence.

## 2. Materials and Methods

### 2.1. Study Design and Sample

The study design and protocols were approved by the local ethics committees in accordance with the Ethical Principles for Medical Research Involving Human Subjects adopted in the Declaration of Helsinki by the World Medical Association and the European and Spanish regulations on the protection of medical data (Institutional Review Board Statement).

This multicentric retrospective cohort study included all patients who underwent first PV isolation procedures with CB or RF techniques for symptomatic paroxysmal or persistent AF, and who were refractory to one or more anti-arrhythmic drugs. Recruitment was performed at different hospitals in the Province of Malaga (Andalusia, Spain) and coordinated by Hospital Universitario Virgen de la Victoria from January 2009 to March 2019.

A total of 1055 patients with a first PV isolation for AF were included in the present study. Patients were divided into two groups according to the ablation technique, the CB (*n* = 557) and RF (*n* = 498) groups. All participants were followed up for 30 months after the index ablation procedure. Those patients with reablation were divided, according to the first ablation procedure, into the CB-redo and RF-redo subgroups and the PV reconnection was analyzed.

### 2.2. CB and RF Ablation Procedures for AF

The CB technique was exclusively a single-shot ablation of the PV ostium with 240-ms applications at −40 °C using 23-mm and 28-mm Arctic Front and Arctic Front Advance catheters (Medtronic, Minneapolis, MN, USA). Phrenic stimulation was performed to monitor the nerve injury. A CT scan was previously performed to assess the anatomy of the PVs.

The RF ablation was conducted with CARTO^®^ (Biosense Webster, Irvine, CA, USA) and NAVX^®^ (Endocardial Solutions, Saint Paul, MN, USA) guidance systems, with the corresponding open-irrigated ablation catheters and updates throughout the duration of the study: ThermoCool^®^ and ThermoCool SmartTouch^®^ (Biosense, Webster, Irvine, CA, USA); Therapy Cool Path^®^ and Therapy Cool Flex^®^ (Abbot, Chicago, IL, USA). The substrate for ablation was selected depending on the activation maps obtained for each patient. Wide antral circumferential ablation (WACA) was performed in all patients aiming for a contiguous circle enclosing the PVs. With contact force-sensing catheters, a real-time automated display of RF applications was used with predefined settings of catheter stability (3 mm for 8 s) and a minimum contact force (30% of time >4 g, between 25–35 W of power) with an irrigation flow up to 30 cm^3^/min. RF was delivered until an ablation index (AI) of ≥400 at the posterior wall/roof and ≥550 at the anterior wall, with an inter-lesion distance of <6 mm. All lesions were performed consecutively without time spacing of the posterior wall lesions. In the case of dislocation, a new RF application reaching the AI target was applied. In the case of no isolation after completing the circumferential ablation, touch-up ablation was delivered until the complete PV isolation. Circumferential PV isolation was the standard procedure, and it was performed on 208 patients. Two hundred and nine patients also received additional lines in the LA roof, and eighty-one patients received both additional lines in the LA roof and in the mitral isthmus. Complete PV isolation was confirmed via the bidirectional entrance and exit block.

### 2.3. Follow-Up

The duration of the follow-up was 30 months from the index ablation procedure, and it was previously established for all patients. Clinical follow-up was scheduled at 3, 6, 12, 18, 24 and 30 months by a cardiologist. At each follow-up time point, patients had an electrocardiogram and a Holter monitor for 24 h or seven days depending on the frequency of symptoms. Anti-arrhythmic therapy was at the discretion of the cardiologist.

All patients received anticoagulant therapy with acenocoumarol or direct oral anticoagulants one month before and three months after the index ablation procedure. After that period, the indication for anticoagulant medication was in accordance with the CHA₂DS₂-VASc score.

### 2.4. Variables and Endpoints

Demographics, comorbidities, clinical data, and echocardiographic and procedural parameters at the time of the index ablation procedure were collected from personal health records. Independent variables associated with the risk of AF recurrence were as follows: age, sex, smoking history, obesity (defined as body mass index >30 kg/m^2^), hypertension, diabetes mellitus, dyslipidemia, AF pattern (paroxysmal or persistent), previous heart disease, left ventricular ejection fraction, previous coronary artery disease and revascularization, presence of dilated left atrium (LA) (diameter in parasternal long-axis view in 2D-transthoracic echocardiography ≥40 mm), early recurrence (detection of AF episodes, flutter or atrial tachycardia lasting more than 30 s in the first 90 days after the index ablation) and the presence of severe obstructive sleep apnea (OSA, defined as >30 on the apnea-hypopnea index) [9,10].

The primary endpoint was the need for a repeat ablation (i.e., reablation) during a 30-month follow-up period after the index ablation procedure with CB or RF technique. Patients selected for reablation were those who presented with recurrence of AF, flutter or atrial tachycardia lasting more than 30 s after 90 days of the index procedure (i.e., blanking period), symptomatic (European Heart Rhythm Association symptom classification (EHRA score) ≥II) and refractory to anti-arrhythmic medication (regimen at the discretion of the cardiologist). The reablation was a final choice in accordance with the patient’s preference. This primary endpoint for reablation was also analyzed for both paroxysmal and persistent AF subsets of patients in the index ablation procedure.

The secondary outcomes were the identification of predictor variables of reablation and the presence of PV reconnection in patients with reablation from the CB group (CB-redo subgroup) and the RF group (RF-redo subgroup). Thus, patients with reablation (CB-redo and RF-redo subgroups) were classified according to the number and location of reconnected PVs.

### 2.5. Statistical Analysis

Qualitative variables are shown as an absolute number and percentage (*n*, %) and continuous variables as a mean and standard deviation (mean ± SD). The differences between the CB and RF groups were analyzed using the Pearson’s Chi-square test or the Fisher’s exact test for qualitative variables, and the Student’s *t*-test or the Mann–Whitney U test for continuous variables. To assess the need for reablation in the CB and RF groups, three Cox regression models with successive adjustments for independent variables were used to estimate the hazard ratios (HR) and 95% confidence intervals (95% CI): (1) age and sex; (2) age, sex, active smoking, hypertension, diabetes mellitus and dyslipidemia; and (3) age, sex, active smoking, hypertension, diabetes mellitus, dyslipidemia and variables associated with reablation with a statistical significance level less than 0.25 (i.e., presence of dilated LA, severe OSA, persistent AF and early recurrence) [11]. In order to identify predictor variables of reablation and to analyze the presence of PV reconnection, another Cox regression model was used. Kaplan–Meier survival curves were used to graphically represent Cox regression models.

All statistical tests were two-sided, and *p*-values less than 0.05 (*p* < 0.05) were considered statistically significant. The statistical analysis was performed with the IBM SPSS Statistics version 21.0 for Windows (Armonk, NY, USA).

## 3. Results

### 3.1. Characteristics of the Sample

The total sample included 1055 patients with AF who underwent a first AF ablation for PV isolation. The sample was composed of 557 patients (52.8%) with CB ablation (CB group), 411 patients for paroxysmal AF (39.0%) and 146 patients for persistent AF (13.8%); and 498 patients (47.2%) with RF ablation (RF group), 349 patients for paroxysmal AF (33.1%) and 149 patients for persistent AF (14.1%).

Table 1 provides demographics, comorbidities and clinical data related to AF of the sample divided into the CB and RF groups. Overall, the patients who underwent AF ablation were males (68.3%) with a mean age of 56.5 ± 11.0 years and a mean BMI of 28.9 kg/m^2^ (overweight), of whom approx. 50% were hypertensive and had dyslipidemia. Although there were no differences between both groups in these comorbidities, the percentage of OSA was significantly higher in the CB group than in the RF group (*p* < 0.05).

More differences were found in previous heart diseases among patients, namely in ischemic cardiomyopathy, hypertensive cardiomyopathy and congenital heart disease. Thus, the CB group had significantly more patients with ischemic cardiomyopathy than the RF group (*p* < 0.05), but significantly fewer patients with hypertensive cardiomyopathy and congenital heart diseases (*p* < 0.05).

Regarding other clinical data related to AF, there were no differences between both groups of patients prior to the index ablation procedure. In the sample, echocardiographic patterns showed preserved global systolic function and a mean diameter of LA of 39.2 mm. In addition, most were found under treatment with beta-blockers (82.5%) and antiarrhythmic drugs (84.5%) at the time of ablation.

### 3.2. Follow-up Period and Reablation

Major and minor complications were reported for CB and RF ablation procedures but there were no significant differences between groups: 2.7% and 4.6% (*p* = 0.099) in the CB group, and 6.1% and 4.0% in the RF group (*p* = 0.161), respectively. However, different adverse events were found depending on the index ablation (Table A1).

All patients were followed up for 30 months after the index ablation procedure. However, 21 patients (2%) were lost during the follow-up period; 10 patients from the CB group (1.8%) and 11 patients from the RF group (2.2%). Early arrhythmia recurrence was observed in 88 patients (8.3%), 50 patients from the CB group (9.0%) and 38 patients from the RF group (7.6%); but the definitive arrhythmia recurrence was found in 301 patients (28.5%); 214 cases were detected within the first 15 months. Among patients with definitive arrhythmia recurrence, 147 patients from the CB group and 154 patients from the RF group, there were no significant differences between both groups (*p* = 0.06). The main arrhythmia recurrence was AF in 87 patients from the CB group (59.2%) and 93 patients from the RF group (60.4%); auricular flutter in 44 patients from the CB group (29.9%) and 49 patients from the RF group (31.8%); and a minority of cases of atrial tachycardia in 16 patients from the CB group (10.8%) and 12 patients from the RF group (7.8%).

Regarding reablation, 223 patients (74.1% of patients with arrhythmia recurrence) were candidates for repeat ablation in accordance with the criteria defined for this study, 85 patients from the CB group (15.3%) and 138 patients from the RF group (27.7%). In this case, the comparison of the need for reablation between both ablation groups revealed a significant difference (*p* < 0.001).

In contrast, 78 patients with arrhythmia recurrence were not eligible for reablation because of the following reasons: EHRA score <II (24 patients (30.7%) from the CB group and 10 patients (11.5%) from the RF group), symptomatic improvement with antiarrhythmic drugs (11 patients (14.1%) from the CB group and 9 patients (11.5%) from the RF group), refusal of repeat ablation (9 patients from the CB group (11.5%) and 13 patients (16.7%) from the RF group) and loss of follow-up (2 patients from the CB group (2.5%) and 1 patient (1.3%) from the RF group). At this point, the composite of symptomatic improvement with antiarrhythmic drugs and the reduction in the EHRA score < II was observed in 35 patients from the CB group (44.9%) and 19 patients from the RF group (24.3%), which resulted in significant differences between groups (*p* = 0.048).

### 3.3. Need for Reablation in Patients with CB and RF Ablation

Table 2 shows the results of three Cox proportional hazard regression models to estimate the effects of covariates on the need for repeat ablation (survival time) based on the index ablation procedure for AF. In the total sample, the Cox regression model 1 adjusted for age and sex showed that patients from the CB group had a significantly lower need for reablation than patients from the RF group (HR = 0.51 and 95% CI = 0.39–0.67; *p* < 0.001). Although the inclusion of cardiovascular risk factors in model 2 had no differential effects on the HR, model 3 showed that the additional adjustment for cardiovascular risk factors and relevant clinical variables decreased even more the need for reablation in patients with CB ablation (HR = 0.45 and 95% CI = 0.32–0.61; *p* < 0.001).

As shown in Figure 1, the Kaplan–Meier curves of the full-adjusted Cox regression model 3 for each group show these differences in the need for reablation: 85 patients in the CB group (15.2%) and 138 patients in the RF group (27.7%).

Given that there were patients with PV isolation and additional ablation lines in the RF index ablation procedure, we assessed the need for repeat ablation during the follow-up using the full-adjusted Cox regression model 3 in subgroups of patients with different RF ablation strategies: 45 patients (32.6%) with circumferential isolation of the PV ostium; 59 patients (42.8%) with circumferential isolation of the PV ostium and additional lines in the LA roof; and 34 patients (24.6%) with circumferential isolation of the PV ostium and additional lines in the LA roof and mitral isthmus. However, there were no significant differences between these RF subgroups (*p* = 0.183).

### 3.4. Need for Reablation in Patients with Paroxysmal and Persistent AF

Because the need for reablation was assessed in patients from the CB and RF groups with arrhythmia recurrence, we distinguished those patients with paroxysmal AF from patients with persistent AF in the index ablation procedure and the primary endpoint during the follow-up period was again assessed using the full adjusted Cox regression model (model 3). The multivariable model in the paroxysmal AF subset of patients revealed that the CB group had a significantly lower need for reablation than the RF group (HR = 0.47 and 95% CI = 0.31–0.70; *p* < 0.001), which is shown in Figure 2A.

Similar to paroxysmal AF, the multivariable model in the subset of patients with persistent AF revealed that the CB group had a significantly lower need for reablation than the RF group (HR = 0.46 and 95% CI = 0.27–0.76; *p* < 0.001), as shown in Figure 2B.

### 3.5. Identification of Predictors of Reablation in Patients with CB and RF Ablation

The covariates included in the full adjusted Cox regression analysis (model 3) were assessed to estimate the relative contribution and the predictive value of each independent variable for reablation in the total sample. Table 3 shows that RF ablation (HR = 2.25 and 95% CI = 1.63–3.10; *p* < 0.001), dilated LA (HR = 1.42 and 95% CI = 1.03–1.96; *p* < 0.05), persistent AF (HR = 1.49 and 95% CI = 91.07–2.06; *p* < 0.05) and early recurrence (HR = 5.85 and 95% CI = 3.94–8.67; *p* < 0.001) were identified as potential predictors of reablation among patients with prior ablation procedure for AF.

### 3.6. PV Reconnection in Patients with Reablation

All patients who underwent reablation were assessed with a three-dimensional mapping of the CB and RF ablation procedures. As previously commented, 85 patients from the CB group and 138 patients from the RF group were treated with repeat ablation (the CB-redo and RF-redo subgroups, respectively). Table A2 shows the demographic and clinical characteristics of these patients prior to the reablation procedure and there were no significant differences between both redo subgroups in the use of beta-blockers (82.4% in the CB-redo subgroup and 62.3% in the RF-redo subgroup), antiarrhythmic drugs (90.6% in the CB-redo subgroup and 94.2% in the RF-redo subgroup) or electrical cardioversion (17.4% in the CB-redo subgroup and 21.2% in the RF-redo subgroup). At the end of the follow-up, 21 patients from the CB-redo subgroup (24.7%) and 33 patients of the RF-redo subgroup (23.9%) needed to continue with additional antiarrhythmic therapy (*p* = 0.101).

As shown in Figure 3A, the percentage of patients with PV reconnection (i.e., at least one reconnected PV) was significantly lower in the CB-redo subgroup (32 patients, 37.6%) than in the RF-redo subgroup (117 patients, 84.7%) as revealed by a Cox regression model adjusted for age, sex, active smoking, hypertension, diabetes mellitus, dyslipidemia, severe OSA, dilated LA, persistent AF and early recurrence (HR = 0.29 and 95% CI = 0.18–0.47; *p* < 0.001). In this multivariable model, the presence of early recurrence was identified as a predictor variable of PV reconnection (HR = 0.16 and 95% CI = 0.09–0.28; *p* < 0.001). Figure 3B shows the number of patients from the CB-redo and RF-redo subgroups with 0, 1, 2, 3 and 4 reconnected PVs, and significant differences were found in the distribution (*p* < 0.001): (i) 53 (62.3%), 22 (25.8%), 6 (7.0%), 4 (4.7%) and 0 (0.0%) patients in the CB-redo subgroup, respectively; (ii) 21 (15.2%), 14 (10.1%), 37 (26.8%), 32 (23.2%) and 34 (24.6%) patients in the RF-redo subgroup, respectively. The mean number of reconnected PVs was significantly lower in the CB-redo subgroup than in the RF-redo subgroup (0.5 ± 1.8 and 2.3 ± 1.4; *p* < 0.001) and the most frequently reconnected PV was the right superior followed by the right inferior PV.

The anatomical region causing the arrhythmia recurrence in these patients is described in Figure 4.

## 4. Discussion

This study assessed the CB and RF ablation procedures in patients with paroxysmal and persistent AF during a 30-month follow-up period. Our results suggest that the CB ablation decreases the need for repeat ablation in arrhythmia recurrence and the percentage of PV reconnection over time in the follow-up in our usual clinical practice. These findings were consistent and were confirmed when multivariable analyses for reablation were adjusted for comorbidities and cardiovascular risk factors, such as severe OSA, dilated LA and early recurrence.

Most patients who experienced arrhythmia recurrence were treated with repeat ablation (74%). Thus, the recurrence rate oscillated between 30 and 50 percent. Because of the difficulty in both definition of recurrence and obtaining documented recurrence, we considered the clinical decision to perform a reablation procedure as the primary endpoint of the study. The need for reablation is more indicative of the real outcome and symptomatic state of patients who underwent a first ablation procedure. This endpoint is clearly reflected in the current guidelines on AF and supported by many published studies, whose main goal is to improve symptoms and quality of life in patients with PV isolation [12,13].

At this point, it is important to indicate that clinically significant recurrence affects the quality of life and led to a reablation. Arrhythmia recurrence depends on several factors, such as the type of first-diagnosed AF, baseline characteristics of patients and even the definition of recurrence. However, the detection of arrhythmia recurrence is also closely related to both the capacity and availability of devices to record events and the speed of detection by emergency services. For instance, a recent study in a similar cohort to ours has shown that the use of implantable electrocardiogram recording devices detects relapse in up to 50 percent of cases [14]. Therefore, we can deduce that there exists a much higher rate of arrhythmia recurrence that it is not clinically detected or remains silent. Given that the indication for AF ablation is essentially clinical, except in the few cases of secondary heart failure [15], the need for reablation provides a great deal of information about the benefits obtained by the patient and constitutes a solid objective endpoint by which to judge the technical success. These findings have also been supported by the present study because there was a significant reduction in the composite of symptomatic improvement with antiarrhythmic drugs and the reduction in the EHRA score after the CB index ablation procedure, although the arrhythmia recurrence in patients from the CB and RF groups was similar. Accordingly, these observations could explain the reduction in the need for repeat ablation with a second procedure.

In terms of the patient profile, the singularity of our study compared to other previously published research consisted of its large sample size, which provided a broad representation of the general population. Moreover, the duration of the follow-up period allowed us to ensure the detection of recurrence and late reablation, thus reducing bias. Both CB and RF ablation procedures were represented by balanced homogeneous groups, which allowed us to attribute the observed differences to each index ablation procedure. The present analysis also included patients with persistent AF, a population that is generally underrepresented in most studies and trials [16], and different strategies of PV isolation in the index ablation, especially in patients with persistent AF. However, the different strategies of PV isolation in the RF group (i.e., circumferential PV isolation, circumferential PV isolation and additional lines in the LA roof, and circumferential PV isolation and additional lines in the LA roof and mitral isthmus) were not associated with differences in the need for reablation during the follow-up period, which is in agreement with previous studies that found no reduction in the rate of arrhythmia recurrence when either linear ablation or ablation of complex fractionated electrograms was performed in addition to PV isolation in the index ablation [3,17].

Despite being a cohort study, these results are in agreement with previous clinical trials, such as both the FIRE AND ICE trial and the FREEZE Cohort Study, which corroborated the equivalent effectiveness of the techniques to maintain sinus rhythm with an 18-month follow-up for paroxysmal AF. The statistical analysis of the results of the FIRE AND ICE trial [18] revealed that CB ablation reduces rehospitalization rates, post-ablation electrical cardioversion and reablation rates, which raises the question of whether the CB recurrence rate might be lower than that of RF. In contrast, other studies found no differences between CB and RF ablation. For example, the multicenter, randomized CIRCA-DOSE study assessed procedures with a short 2-min CB ablation, standard 4-min CB ablation and RF ablation in patients with drug-refractory paroxysmal AF, but there were no differences after a 12-month follow-up between techniques in the rate of repeat ablation or PV reconnection. Unlike our study, all patients with RF ablation in the CIRCA-DOSE study used catheters with contact-force technology [19]. This technology allowed us to reach a minimum ablation index (i.e., weighted formula that includes contact force, time and power) of 400 for each lesion, which could explain these differences. However, according to Fire and Ice investigators, in cases of unstable catheters, an increased power could not produce an appropriate lesion that leads complete isolation of the PV ostia [20].

As a secondary endpoint, we found that the probability of PV reconnection during the follow-up was significantly lower in patients who underwent CB ablation for AF than in those patients with RF ablation, adjusted for possible confounding factors. Consequently, the number of reconnected PVs was significantly less among patients from the CB-redo subgroup and a large proportion of patients (62.3%) had the PVs isolated. This leads us to propose that the lesions in the PV ostia produced by CB may be longer-lasting and more stable over time compared with RF. However, in some cases, the arrhythmogenic substrate of the recurrence was not only focused on the PV ostia, but on other more complex and extensive circuits that coexist. This is probably associated with a higher fibrotic load in the left atrium that requires more complex ablations (additional lines, ablation of the posterior wall, ablation of the mitral isthmus). In such cases, the capacity to fully eradicate the arrhythmogenic substrate is more limited, regardless of the technique used. Moreover, the only predictor variable of PV reconnection during the follow-up was the presence of early recurrence.

In terms of safety, there were no significant differences in the rate of complications between patients with CB and RF ablation procedures, although the types of complications were different, as previously mentioned. The present results are comparable to those reported in previous clinical research, with rates of major and minor complications less than 2 and 8%, respectively [21].

Our results suggest that CB ablation could be used as the technique of choice for the first ablation procedure in patients with AF. Because great benefits derived from CB have been described in the literature (e.g., short intervention time and short learning curve) [22], it is feasible that this technique could be used in a larger number of hospitals rather than being limited to specialized institutions. This would imply wider coverage of the patients with AF, a population that is experiencing exponential growth and increasingly demands more effective and faster attention.

Delays in the treatment of AF and ablation procedures are associated with worse outcomes and favor the progression of arrhythmias. Therefore, these data should be considered when deciding the protocols and techniques to use as the first procedure, bearing in mind that there is no consensus.

To conclude, there is a tendency to use RF for second ablation procedures because it is more flexible for the evaluation and treatment of PVs (i.e., PV reconnection), as well as for the analysis of new arrhythmogenic substrates or placement of the necessary ablation lines. However, several observational studies and one randomized study found no significant differences in arrhythmia recurrence after repeat ablation with CB and RF procedures [23,24]. Therefore, it is possible that CB is currently an underused technique for repeat ablation procedures.

### Limitations

We are aware of several limitations in this study. First, this is a retrospective, non-randomized cohort study that included multivariable analyses with several independent variables (covariates) from medical records and clinical examinations; however, there are confounding variables that were not assessed but could be relevant for the endpoints. Second, this is a multicenter study conducted in the south of Europe (Málaga, Andalusia, Spain) but there are several dietary and environmental factors (e.g., pollution, cultural models of lifestyle, healthcare systems) that can influence the outcomes. Third, the improvement of the catheter technology over time, with the later inclusion of contact-force in RF or the development of new CB catheters, and the application of new protocols, especially in RF ablation (i.e., the high-power short-duration index-guided ablation (HPSD) protocol) are factors that may affect results in this retrospective study, but this is something inherent to the scientific and technological advances of recent years. Fourth, it is important to highlight that different ablation strategies were performed within the RF group, according to the activation maps obtained in each patient. However, this heterogeneity in the index ablation of patients from the RF group was not associated with differences in the need for repeat ablation during the follow-up period. Fifth, the reablation procedure was performed taking into account the patient’s preferences, which entails subjectivity and a lack of clinical criteria; in fact, in a few cases, some patients decided not to undergo a reablation.

Despite these limitations, we have included consecutive patients with AF who underwent PV isolation using both CB and RF ablation procedures in our clinical practice, unlike other studies that only included patients selected from clinical trials.

## 5. Conclusions

In conclusion, our study suggests that CB ablation reduces the need for repeat ablation in patients with paroxysmal or persistent AF and reduces the PV reconnection during a long-term follow-up period of 30 months in comparison with the RF ablation procedure. Therefore, these results could be considered before planning a first ablation procedure in patients with AF. Future prospective randomized trials are needed to compare both ablation procedures based on the freedom of reablation and the presence of the PV reconnection after the first ablation.

## Figures and Tables

**Figure 1 jcm-11-05862-f001:**
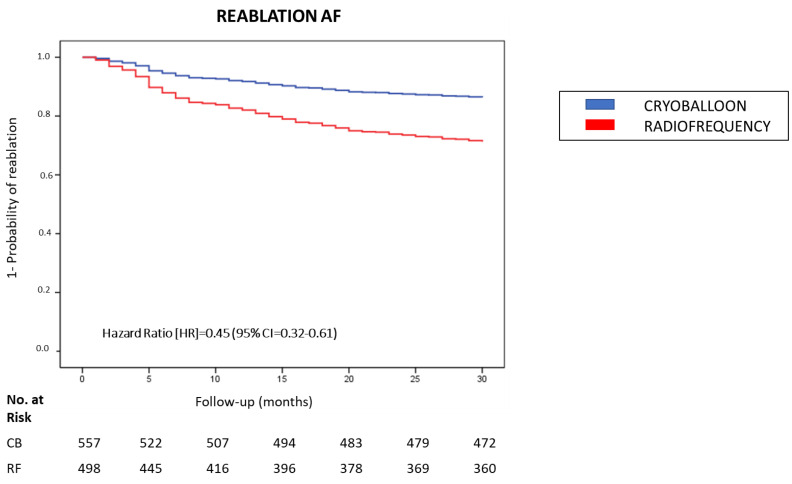
The need for reablation in AF in patients with CB and RF ablation. Kaplan–Meier curves of the need for reablation during a 30-month follow-up based on the procedure for AF with CB or RF ablation using a Cox regression model adjusted for: age, sex, active smoking, hypertension, diabetes mellitus, dyslipidemia, severe OSA, dilated LA, persistent AF and early recurrence. Abbreviations: AF = atrial fibrillation; CB = cryoballoon; CI = Confidence Interval; HR = hazard ratio; LA = left atrium; OSA = obstructive sleep apnea; RF = radiofrequency.

**Figure 2 jcm-11-05862-f002:**
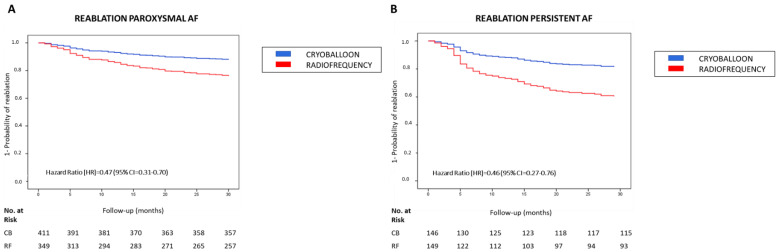
The need for reablation in patients with CB and RF ablation procedures for paroxysmal or persistent AF. Kaplan–Meier curves of the need for reablation during a 30-month follow-up based on the procedure for AF with CB or RF ablation using Cox regression models adjusted for age, sex, active smoking, hypertension, diabetes mellitus, dyslipidemia, severe OSA, dilated LA, persistent AF and early recurrence. (**A**) The need for reablation in paroxysmal AF; (**B**) The need for reablation in persistent AF. Abbreviations: AF = atrial fibrillation; CB = cryoballoon; CI = Confidence Interval; HR = hazard ratio; LA = left atrium; OSA = obstructive sleep apnea; RF = radiofrequency.

**Figure 3 jcm-11-05862-f003:**
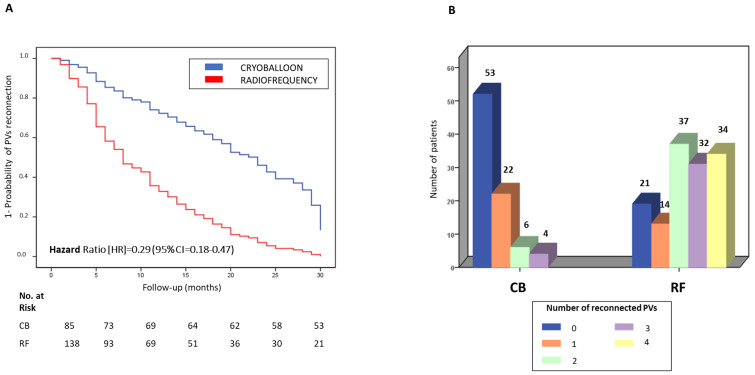
PV reconnection in the reablation subgroups from patients with CB and RF ablation procedures. Evidence of PV reconnection and number of reconnected PVs during a 30-month follow-up period in patients from the CB-redo and RF-redo subgroups. (**A**) Kaplan–Meier curves of the PV reconnection in reablation in the CB-redo or RF-redo subgroup using a Cox regression model adjusted for: age, sex, active smoking, hypertension, diabetes mellitus, dyslipidemia, severe OSA, dilated LA, persistent AF and early recurrence; (**B**) Number of patients with 0, 1, 2, 3 and 4 reconnected PVs from the CB-redo and the RF-redo subgroups. Abbreviations: AF = atrial fibrillation; CB = cryoballoon; CI = Confidence Interval; HR = hazard ratio; LA = left atrium; OSA = obstructive sleep apnea; PV = pulmonary vein; RF = radiofrequency.

**Figure 4 jcm-11-05862-f004:**
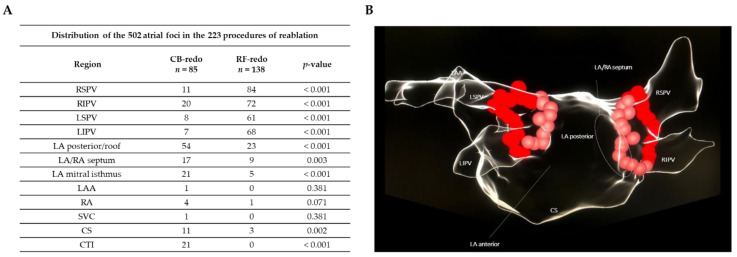
Exact distribution in the PVs of the 502 atrial foci from the redo subgroups observed in the 223 procedures of reablation. (**A**) Anatomical distribution in the LA and PVs of the atrial foci causing the arrhythmia recurrence in patients from the CB-redo and RF-redo subgroups. (**B**) Three-dimensional diagram of the anatomy of the LA and PVs. Abbreviations: CB = cryoballoon; CS = coronary sinus; CTI = cavotricuspid isthmus; LA = left atrium; LAA = left atrial appendage; LIPV = left inferior pulmonary vein; LSPV = left superior pulmonary vein; PA = posterior-anterior; PV = pulmonary vein; RIPV = right inferior pulmonary vein; RSPV = right superior pulmonary vein; RA = right atrium; RF = radiofrequency; SVC = superior vena cava.

**Table 1 jcm-11-05862-t001:** Baseline characteristics of patients with CB and RF ablation procedures.

Characteristic	Total *n* = 1055	CB Group *n* = 557	RF Group *n* = 498	*p*-Value
Age (years); mean ± SD	56.5 ± 11.0	57.0 ± 11.0	55.8 ± 11.0	0.103
Sex, male; *n* %	721 (68.3%)	376 (67.5%)	345 (69.3%)	0.291
Body mass index (kg/m^2^); mean ± SD	28.9 ± 4.9	29.3 ± 5.0	28.4 ± 4.6	0.105
Active smoking; *n* %	133 (15.6%)	72 (15.9%)	61 (15.3%)	0.276
Hypertension; *n* %	589 (56.9%)	322 (57.9%)	267 (55.7%)	0.261
Diabetes mellitus; *n* %	123 (11.9%)	60 (10.8%)	63 (13.2%)	0.141
Dyslipidemia; *n* %	370 (42.6%)	199 (43.4%)	171 (41.8%)	0.348
Severe OSA; *n* %	233 (26.1%)	140 (28.7%)	93 (22.9%)	0.027
LA diameter (mm); mean ± SD	39.2 ± 4.2	38.6 ± 3.8	39.8 ± 4.4	0.225
Previous heart disease(s); *n* %	156 (14.8%)	86 (15.4%)	70 (14%)	0.374
- Dilated cardiomyopathy	13 (1.2%)	9 (1.6%)	4 (0.8%)	0.181
- Ischemic cardiomyopathy	65 (6.1%)	42 (7.5%)	23 (4.6%)	0.032
- Hypertensive cardiomyopathy	17 (1.6%)	5 (0.9%)	12 (2.4%)	0.044
- Hypertrophic cardiomyopathy	12 (1.1%)	4 (0.7%)	8 (1.4%)	0.143
- Tachymyocardiopathy	18 (1.7%)	12 (2.1%)	6 (1.2%)	0.171
- Valvular heart disease	26 (2.4%)	13 (2.3%)	13 (2.6%)	0.463
- Alcoholic cardiomyopathy	1 (0.1%)	1 (0.2%)	0 (0.0%)	0.528
- Congenital heart disease	4 (0.4%)	0 (0.0%)	4 (0.8%)	0.049
Previous PCI; *n* %	43 (4.1%)	27 (4.8%)	16 (3.2%)	0.164
Previous CABG; *n* %	2 (0.2%)	2 (0.4%)	0 (0.0%)	0.110
Left systolic ventricular function (%); mean ± SD	59.0 ± 4.8	58.5 ± 5.9	59.4 ± 3.1	0.288
Previous treatment; n %				
- Beta-blockers	870 (82.5%)	444 (79.7%)	426 (85.5%)	0.113
- Antiarrhythmic drugs	892 (84.5%)	469 (84.3%)	423 (84.9%)	0.521
Persistent AF; *n* %	295 (28.0%)	146 (26.2%)	149 (29.9%)	0.102

Data shown as mean and standard deviation (mean ± SD) or absolute number and percentage (*n* %). Abbreviations: AF = atrial fibrillation; CABG = coronary artery bypass grafting; LA = left atrium; OSA = obstructive sleep apnea; PCI = percutaneous coronary intervention; SD = standard deviation

**Table 2 jcm-11-05862-t002:** Cox regression models adjusted for different independent variables to assess the need for reablation in patients with CB and RF ablation procedures.

Cox Regression Model	Independent Variable	HR (95% CI)	*p*-Value
Model 1	Age	0.51 (0.39–0.67)	<0.001
	Sex		
Model 2	Age	0.47 (0.36–0.60)	<0.001
	Sex		
	Active smoking		
	Hypertension		
	Diabetes mellitus		
	Dyslipidemia		
Model 3	Age	0.45 (0.32–0.51)	<0.001
	Sex		
	Active smoking		
	Hypertension		
	Diabetes mellitus		
	Dyslipidemia		
	Severe OSA		
	Dilated LA		
	Persistent AF		
	Early recurrence		

Abbreviations: AF = atrial fibrillation; CI = Confidence Interval; HR = hazard ratio; LA = left atrium; OSA = obstructive sleep apnea.

**Table 3 jcm-11-05862-t003:** Identification of predictor variables of reablation during a 30-month follow-up period in patients with ablation for AF.

Independent Variable	Cox Regression Model for Reablation		
	HR	95% CI	*p*-Value
RF ablation	2.25	1.63–3.10	<0.001
Age	0.99	0.97–1.00	0.161
Sex, male	1.12	0.79–1.59	0.526
Active smoking	0.79	0.50–1.16	0.203
Hypertension	1.12	0.80–1.58	0.513
Diabetes mellitus	1.02	0.66–1.61	0.927
Dyslipidemia	1.07	0.80–1.41	0.659
Severe OSA	1.36	0.98–1.90	0.067
Dilated LA	1.42	1.03–1.96	0.032
Persistent AF	1.49	1.07–2.06	0.017
Early recurrence	5.85	3.94–8.67	<0.001

Abbreviations: AF = atrial fibrillation; CI = Confidence Interval; HR = hazard ratio; LA = left atrium; OSA = obstructive sleep apnea; RF = radiofrequency.

## Data Availability

The data presented in this study are available on request from the corresponding authors. The data are not publicly available due to ethical and privacy restrictions.

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
