# Peer review of "Reablation in Atrial Fibrillation Recurrence and Pulmonary Vein Reconnection: Cryoballoon versus Radiofrequency as Index Ablation Procedures"

_jcm, 2022, doi:10.3390/jcm11195862_

Round 1
Reviewer 1 Report
Molina-Ramos et al. reported a large retrospective cohort study conducted in >1000 patients who underwent a first AF ablation. Their aim was to assess the need for reablation among pts who underwent CB or RF. The study is well written and clear. Please find enclosed some comments:
First, the title is not clear. I would define better the idea of an index procedure done using CB vs. RF and not just saying "CB or RF".
Second, it would be of interest to define better the RF quality lesion. Please, report whether LSI- or AI-guidance was used and if, the prespecified values. Define if WACA or segmental ablation has been performed. Since the follow-up was up to 30 months, I assume that no pts has been treated using HPSD protocol. This needs to be stated the limitations.
Third, it would be of great interest to know the PVs region of reconnection and not just how many PVs.
Reviewer 2 Report
First of all thank you for giving me the chance to review this manuscript. Molina-Ramos et al sought to compare patients who underwent CB and RF ablation procedures for AF in terms of reablation. In addition, predictors of reablation were assessed through the Cox regression method.
The following issues should be addressed:
- There is a huge amount of patients in the RF group which underwent ablation of additional lines (209 roof line and 81 both roof and mitral line). So, the comparison is between PV isolation vs an heterogeneous group including PV isolation/roof line/roof line+mitral line. This might have affected the results and should be stated and discussed better.
- It would be interesting to also know the actual rate of AF recurrence in both group in order to understand whether the reduction in reablations reflects a real reduction in AF recurrence or is mostly related to a reduction of AF burden or to an improvement of quality of life
- Please specify the complications: in the Table global complications are 9.2% in the CB group and 8.8% in the RF group. This rate is quite high. What do you mean for global complications? I suggest to only report major complications or, alternatively, to divide major and minor complications specifying the type of complication
- Is it possible to specify the type of recurrence for each group? (atrial tachycardia , atrial flutters, AF)
- Can you please state how many patients for both groups were still on antiarrhythmic drugs?
